# Getting to the Emergency Department in time: Interviews with patients and their caregivers on the challenges to emergency care utilization in rural Uganda

**Ashley E. Pickering**[1,2☯]*, **Heather M. Dreifuss**[3☯], **Charles Ndyamwijuka**[2☯], **Mark Nichter**[4,5☯], **Bradley A. Dreifuss**[2,5,6☯], **on behalf of the Global Emergency Care Collaborative Investigators**[¶]

**1** Department of Emergency Medicine, University of Maryland Medical Center, Baltimore, MD, United States of America, **2** Global Emergency Care, Shrewsbury, MA, United States of America, **3** Department of Health Sciences, Northern Arizona University, Flagstaff, AZ, United States of America, **4** School of Anthropology, University of Arizona, Tucson, AZ, United States of America, **5** Mel and Enid Zuckerman College of Public Health, University of Arizona, Tucson, AZ, United States of America, **6** Department of Emergency Medicine, College of Medicine, University of Arizona, Tucson, AZ, United States of America

☯ These authors contributed equally to this work.
¶ Membership of the Global Emergency Care Collaborative Investigators is provided in the Acknowledgments.
* AshleyPickering@gmail.com

**Data Availability Statement:** Our IRB does not allow for data sharing, so we will be unable to

## Abstract

### Objectives

Karoli Lwanga Hospital and Global Emergency Care, a 501(c)(3) nongovernmental organization, operate an Emergency Department (ED) in Uganda's rural Rukungiri District. Despite available emergency care (EC), preventable death and disability persist due to delayed patient presentations. This study seeks to understand the emergency care seeking behavior of community members utilizing the established ED.

### Methods

We purposefully sampled and interviewed patients and caregivers presenting to the ED more than 12 hours after onset of chief complaint in January-March 2017 to include various ages, genders, and complaints. Semistructured interviews addressing actions taken before seeking EC and delays to presentation once the need for EC was recognized were conducted until a diverse sample and theoretical saturation were obtained. An interdisciplinary and multicultural research team conducted thematic analysis based on descriptive phenomenology.

### Results

The 50 ED patients for whom care was sought (mean age 33) had approximately even distribution of gender, as well as occupation (none, subsistence farmers and small business owner). Interviews were conducted with 13 ED patients and 37 caregivers, on the behalf of

publish the full interview transcripts (our data set). However, as suggested we have added a supplement including our code book with further representative quotes to ensure readers have as broad an access as ethically possible. Data cannot be shared publicly because of this is not allowed based on the IRB for this study. Data are available from the The AIDS Support Organization (TASO) Institutional Data Access / Ethics Committee (contact via phone at: +256 414 532 580, or email: mail@tasouganda.org) for researchers who meet the criteria for access to confidential data.

**Funding:** The author(s) received no specific funding for this work.

**Competing interests:** The authors have declared that no competing interests exist.

patients when unavailable. The median duration of patients' chief complaint on ED presentation was 5.5 days. On average, participants identified severe symptoms necessitating EC 1 day before presentation. Four themes of treatment delay before and after severity were recognized were identified: 1) Cultural factors and limited knowledge of emergency signs and initial actions to take; 2) Use of local health facilities despite perception of inadequate services; 3) Lack of resources to cover the anticipated cost of obtaining EC; 4) Inadequate transportation options.

## Conclusions

Interventions are warranted to address each of the four major reasons for treatment delay. The next stage of formative research will generate intervention strategies and assess the opportunities and challenges to implementation with community and health system stakeholders.

## Introduction

In low- and middle-income countries (LMICs), 24 million lives are lost annually due to emergency conditions that could be addressed through implementation of comprehensive emergency care (EC) systems, with 21% of these deaths occurring in sub-Saharan Africa (SSA) [1]. In 2007, the World Health Assembly resolution on EC systems encouraged health systems research enabling the expansion of effective EC systems globally. More specifically, it urged member states to identify the unmet needs of populations, institute formal EC systems, and develop capacity as a means to reduce catastrophic morbidity and mortality [2]. In 2019, World Health Assembly resolution 72.16 redoubled and expanded these calls to action, including encouraging initiatives to increase awareness of, and expand management capacity for, emergency situations at the community level [3]. Despite the recent establishment of academic EC training and clinical care programs in SSA, there has been limited large-scale implementation of community-based EC services.

Until recently, Uganda, like many LMICs lacked EC services. Efforts are under way to develop a well-coordinated EC system, train appropriate workforce to staff EC facilities, and inform the community about new EC resources available. As part of these efforts, in 2008, Global Emergency Care (GEC), a US-based 501(c) [3] nongovernmental organization, partnered with Karoli Lwanga Hospital, known locally as Nyakibale, to build and staff rural Uganda's first Emergency Department (ED) with nonrotating, formally trained staff. This ED provides high-quality, timely care, evaluating and stabilizing patients before requiring payment. However, patients often arrive at the ED late in the course of illness or injury, reducing the potential impact of EC interventions.

Delays in utilization of available EC are widely recognized though not well understood, ranking second among 7 priority research questions identified to facilitate the improvement of EC in LMICs through a research priority setting exercise conducted by a multidisciplinary coalition of researchers and stakeholders, the Global Emergency Care Research Network [4]. The global emergency medicine community recognizes the need for qualitative research to better understand the perspectives and experiences of those seeking to utilize EC, as well as assess the socioeconomic, cultural, and structural barriers leading to treatment delay [5]. Our

research goal was to better understand these barriers, particularly the challenges rural Ugandans face in accessing and utilizing EC, by interviewing patients and caregivers in the ED.

## The Rukungiri District

The Rukungiri District is located in rural Southwest Uganda. The district is mainly farmland with one urban center. Dirt roads traverse steep, rocky hills [6]. There are 2 "rainy seasons," when heavy rains render dirt roads barely passable for days. Motorcycle taxis are the most prevalent mode of vehicular transportation, as they can navigate footpaths reaching homes and small villages. As of the last published district-level census in 2014, very few households owned automobiles; 7% of households owned motorcycles and 19%, bicycles [7].

The vast majority of inhabitants are subsistence farmers (78%), and most lack regular access to money, especially between harvest seasons [6]. Most homes have no electricity [7]. In 2019, there were 57 mobile phone subscriptions per 100 Ugandans nationwide [8]. While mobile phone ownership is increasing substantially, credit is typically purchased only as needed and may not be available at the time of an emergency.

## The Ugandan health care system

Like much of SSA, the Rukungiri District has a pluralistic health care delivery model comprising:

- "Traditional" and allopathic practitioners

- Private, nonprofit, and government health facilities

- Vendors selling medicines in small shops and local markets

Traditional healers are commonly consulted, and one source estimated that 90% of rural Ugandans utilize traditional remedies [9–11]. In many small villages, private clinics and pharmacies are often staffed by laypersons with no formal health care training. In major trading centers, private clinics staffed by nurses, and possibly a physician, may have limited laboratory and/or imaging capacity. The quality of care at private health facilities varies widely, and their patients generally must pay out of pocket before receiving services.

The Ugandan government provides a multitiered health care system with the health service infrastructure following a pattern of the country's administrative units, with health centers (HCs) of increasing capacity (designated levels II–IV), general hospitals (formerly known as district hospitals), and regional referral hospitals. HC IIs are nurse-run clinics at the village level, while larger facilities (HC III and IV) in trading centers are staffed by midlevel providers, typically Clinical Officers with 2–3 years of medical training, and deliver primary care and maternity services, with capacity to provide limited surgical, inpatient, laboratory, and blood transfusion services [12, 13]. User fees for government health care were abolished in 2001. Following this, an initial decline in services and drug shortages increased the use of private health care [14]. In 2017, when we conducted these interviews, there was no ambulance available in the district.

## Karoli Lwanga Hospital ED

Karoli Lwanga Hospital (Nyakibale) is a private not-for-profit district hospital. In 2008, GEC partnered with the hospital to open the first fully functional ED in Uganda and a 2-year Emergency Care Practitioner (ECP) training program for Ugandan nurses. The ECP program was developed to address Uganda's chronic physician shortage (3 physicians per 100 000 residents in the Rukungiri District as of 2015) and lack of specialty trained EC professionals [6]. The

program's goal was to train ED-based nonphysician clinicians to triage, evaluate, stabilize, and treat patients before admitting them to already-overburdened hospital wards, sending them for surgery, or discharging them home. Unlike other Ugandan health care facilities, patients are clinically stabilized prior to payment, and the ED treats any patient who presents [15]. However, acuity is high, with ED patient mortality of 2.1% and an admission rate of 66.3% (2017, unpublished data). Delayed ED presentation can reasonably be surmised to be among the causative factors.

## Methods

To better understand the challenges rural Ugandans face in accessing and utilizing EC we purposefully sampled and interviewed patients presenting to the ED more than 12 hours after onset of chief complaint, or their caregiver when the patient themselves was unavailable. Semi-structured interviews addressed actions taken before seeking EC and delays once the need for EC was recognized. We purposefully interviewed a diverse sample of patients by demographics and chief complaints. An interdisciplinary and multicultural research team conducted thematic analysis based on descriptive phenomenology using the Framework Method.

### Participant selection

Semistructured interviews with ED patients or their caregivers were conducted at Karoli Lwanga Hospital. A purposeful sample of 50 patients arriving to the ED ≥12 hours after onset of the complaint that they sought care for (chief complaint (CC)) were interviewed between January 19 and March 29, 2017. ≥12 hours was set as a benchmark based on advice from the Ugandan clinicians working in the ED due to their observation that patients arriving outside of this timeframe are subjectively noted to have poor outcomes attributable, at least in part, to delayed presentation. The sample was selected to take into account age, gender, type and severity of CC, as well as "information-rich cases," which we defined as individuals having significant experience with challenges in EC access and utilization, based on conversations with ED staff [16]. These conversations often focused on challenges with identifying the need for EC, obtaining EC from local facilities, finance, or transport. Further, as weather may influence delay, the interviews spanned dry and rainy seasons. For patients with known chronic disease, we included in our sample only those with acute presentations of the disease, rather than those simply seeking ongoing treatment related to the disease. Patients or their caregiver were approached in person in the ED or hospital wards. Recruitment and interviews did not interfere with patients' clinical care and were not conducted when the patient was severely ill but rather once they were stabilized, if at all. The "caregiver" accompanying the patient, was interviewed with pediatric patients and patients who could not be interviewed directly because of illness or preference.

### Ethics

Research ethics approval was obtained through the University of Arizona Institutional Review Board (Tucson), The AIDS Support Organization Research Ethics Committee (Kampala, Uganda), and the Ugandan National Council of Science and Technology (Kampala). Karoli Lwanga Hospital leadership supported the study. Verbal informed consent was approved by the organizations listed above due to low literacy rates in the community. It was obtained from all participants, witnessed by the lead researcher (AP) and head research assistant (NC) and documented on a verbal consent form with AP's signature. In the case of caregiver interviews, assent was obtained from patients old enough, or alert enough to provide it. In most instances this applied to the young children, the elderly and those with severe illness.

## Interviews

Each interview included 10 core questions and appropriate follow-up queries probing actions taken before seeking EC and delays to presentation once the need for EC was recognized. The interview guide was developed by AP (graduate training in public health, qualitative methods and medical anthropology), BD (emergency medicine global health fellowship and graduate training in public health), MN (Regents Professor of medical anthropology), and HD (graduate training in public health and qualitative research experience) in consultation with Ugandan members of the GEC Research Team. It was based on the guiding theory of the Three Delays Model [17], described in depth in the Discussion, and on prior quantitative data regarding barriers to ED utilization at Karoli Lwanga Hospital ED (unpublished). The interview guide remained constant, with minor changes to wording throughout the 50 interviews. The lead researcher (AP) and Ugandan members of the GEC Research Team translated the core questions into the local dialects (Runyankore/Rukiga). Once consensus was reached in the phrasing of questions, back-translation to English was undertaken. The guide was initially piloted with community members not seeking EC followed by approximately ten ED patients or their caregivers. The English version is provided as S1 File.

After didactic training in qualitative interview-based research methods by AP, the head research assistant (NC), a resident of Rukungiri District and native Runyankore/Rukiga and fluent English speaker, obtained practical training, experience and feedback while piloting the interview guide. NC conducted all interviews with AP present. AP was not involved in providing clinical care. NC, a Ugandan community member not involved in clinical care, approached patients to gauge interest in study participation, limiting pressure to participate. Interviews, including only the participants and these 2 researchers, took place in a private room outside of the ED to ensure confidentiality and lasted 15–60 minutes. Interviews were conducted in local dialects, English, or a combination, based on participant preference. Interviews were audio recorded, transcribed verbatim, and translated into English within one week of the interview by NC, then checked for accuracy by a second bilingual member of the GEC Research Team (KA). If there was a discrepancy in translation/interpretation the audio recording was reviewed by NC and KA together and discussed until consensus was reached. Due to the nature of the sample—patients utilizing the ED—typically participants were not available on site when transcription and translation were complete. The IRB limited collection of contact information to ensure privacy, therefore transcripts were not returned to interviewees for review.

## Analysis

An interdisciplinary, multicultural data analysis team composed of ECPs, Ugandan members of the GEC Research Team, and AP was assembled to reduce the possible bias of a single investigator's interpretation and to enhance the cultural depth of the data analysis [18–21]. The Framework Method, a rigorous and transparent process that produces a clear audit trail of each step of data analysis carried out by multidisciplinary research teams, was utilized [19, 21]. As little is known about delays in accessing available EC, in 2017 when the study was conducted, a thematic analysis based on descriptive phenomenology was utilized. The data analysis team met weekly, utilizing the Framework Method to inductively identify:

- Concepts of delay present in multiple de-identified interviews (subthemes)

- Quotes depicting the subthemes

- Quotes depicting interesting outliers

After the first 25 interviews were conducted and analyzed, the data analysis team identified that no additional large-scale concepts of delay were emerging; thematic saturation was reached. The team convened to review subthemes across these first 25 interviews simultaneously and organize them, yielding the 4 themes described in Results. These themes and subthemes constituted a codebook, included as S2 File, facilitating coding of an additional 25 interviews completed to capture a broader sample of demographic groups and chief complaints across both dry and rainy seasons, given differences in EC accessibility and subsistence agriculture activities. AP and NC then coded the identified themes, subthemes, and quotes in the transcripts using QSR International's NVivo software, version 10, which was also used to analyze themes of delay by patient and interviewee demographics. The Consolidated Criteria for Reporting Qualitative Research (COREQ) was utilized to ensure comprehensive reporting [22].

## Results

### Demographics

The 50 ED patients for whom care was sought (13 of whom were interviewed directly) had a mean age of 33 years. Patients had an approximately even distribution of gender, as well as occupation (none, subsistence farmers and small business owner). Demographics and CCs for the ED patients are illustrated in Table 1. A full list of CCs is included as S3 File. Fifty interviews were completed, 37 with caregivers of ED patients and 13 with the ED patients themselves. Interviewees had a mean age of 38 years, and the majority were women, worked as subsistence farmers, and had received primary education. The demographics of interview participants are presented in Table 2.

**Table 1. Patients' demographic data and CC.**

| | |
|---|---|
| **Gender, no. (%)** | Male: 27 (54) |
| | Female: 23 (46) |
| **Age (years)** | Mean: 33.2 |
| **Intervals, no. (%)** | < 1: 5 (10) |
| | 1–4: 5 (10) |
| | 5–17: 7 (14) |
| | 18–55: 22 (44) |
| | > 55: 11 (22) |
| **Occupations (Common), no. (%)** | None: 12 (30) |
| | Subsistence farmer: 16 (32) |
| | Employed or business owner: 13 (26) |
| **Highest level of education** | Median: Primary (n = 20; 40%) |
| | Range: None-University |

| CC by body system and/or mechanism of injury, no. (%)* | |
|---|---|
| Medical illness: 38 (76) | Injury: 12 (24) |
| Gastrointestinal/genitourinary: 12 (32) | Blunt trauma/fall: 5 (42) |
| Other medical complaints: 10 (26) | Burn: 3 (25) |
| Cardiopulmonary: 9 (24) | Road traffic accident: 3 (25) |
| Neurologic: 7 (18) | Bite: 1 (8) |

*See S3 File for full list of CCs.

Abbreviations: CC, chief complaint.

**Table 2. Interview participants' demographic data.**

| Relationship to patient, no. (%) | Parent: 19 (38) |
|---|---|
| | Self:13 (26)* |
| | Adult Child: 9 (18) |
| | Other family member: 9 (18)** |
| **Gender, no. (%)** | Female: 36 (72) |
| | Male: 14 (28) |
| **Age (years)** | Mean: 38.1 |
| | Range: 20–82 |
| **Occupation, no. (%)** | Subsistence farmer: 33 (66) |
| | Employed or business owner: 14 (28) |
| | Student: 3 (6) |
| **Highest level of education** | Median: Primary (n = 32; 64%) |
| | Range: None-University |

* Patient interviewed directly.

** Other family member including grandchildren, siblings, cousins, daughter or son in-law, nieces, and nephews.

## Delay in ED presentation

All patients included in the study reported the acute onset of new symptoms or acute worsening of previously recognized symptoms leading to ED presentation. Delays were quantified as the duration of the patient's CC and the time from recognition of a severe condition necessitating EC (by the patient or their caregiver) to presentation at the ED. Both are presented in Table 3. The most notable trend related to treatment delay at the ED was history of chronic disease. ED patients with acute presentations and history of underlying chronic conditions experienced substantially longer delays in reaching the ED than otherwise healthy patients. Among chronically ill patients, recognition of acute symptoms needing EC ranged from 12 hours– 14 days prior to arriving at the ED, with an average of 6 days. For all patients included in the study the mean duration from recognition of symptoms necessitating EC to presentation at the ED as 1 day. Patients with chronic disease and their caregivers reported challenges to decision-making including perceptions that:

- acute decompensation of chronic disease may be self-limited

- the patient's medication regimen for chronic illness would also treat acute decompensation

- the incurable nature of the disease meant acute stabilization and treatment of complications was futile and therefore a waste of limited resources

No other major trends in length of delay were identified based on patient demographics, or medical illness vs injury.

**Table 3. Patients' delays in ED presentation.**

| | All patients | Medical Illness | History of Chronic Disease | Injury |
|---|---|---|---|---|
| Duration of CC | Median: 5.5 d | Median: 7 d | Median: 112 d | Median: 2 d |
| | Range: 12 h-3 y | Range: 1 d-3 y | Range: 7 d-3 y | Range: 12 h-30 d |
| Time from recognition of need for EC to ED presentation | Median: 1 d | Median: 1 d | Median: 6 d | Median: 1.5 d |
| | Range: 2 h-17 d | Range: < 1 h-17 d | Range: 12 h-14 d | Range: < 1 h-7 d |

Abbreviations: CC, chief complaint; d, day; EC, emergency care; ED, Emergency Department; h, hour; y, year.

**Table 4. Themes and subthemes in delay identified.**

| Themes | Subthemes |
|---|---|
| **Cultural Factors and Limited Knowledge of Emergency Signs and Initial Actions to Take** | Difficultly recognizing "danger signs"<br>• Patient's condition becoming severe before identifying the need for and/or seeking EC<br>• Perception of "cultural illness" |
| **Use of Local Facilities for Stabilization and Advice Despite Perception of Inadequate Services** | • Limited first aid or home care<br>• Use of local facilities:<br>1. Assessable on foot or bicycle<br>2. For advice and symptom management<br>3. While gathering funds needed for EC<br>4. Despite perception of poor care at high cost<br>• Use of multiple facilities before seeking EC<br>• Perception of delays in referral to EC |
| **Lack of Resources to Cover the Direct, Indirect, and Opportunity Costs of EC** | • Perception of EC as unaffordable<br>• Need for funds for transport prior to seeking EC<br>• Opportunity costs of EC |
| **Inadequate Transportation Options, Especially at Night and in Inclement Weather** | • Inadequate vehicles available for the patient's condition<br>• Dirt roads virtually unpassable in rain<br>• Safety concerns when traveling at night |

## Themes in delay

Delay in ED presentation provides the quantitative explanation of delays, whereas themes in delay—presented below—seek to further understand and explain these delays in ED presentation. An overview of the themes and subthemes of delay identified is presented in Table 4.

**Cultural factors and limited knowledge of emergency signs and initial actions to take.**
Approximately half of study participants reported some specific symptom, such as confusion or malaise, as a "danger sign" and increasing symptom severity as a primary rationale for their decision to seek health care. Participants also reported that they did not consider utilizing EC until they feared death was imminent.

> On Saturday night around 2:00 AM the symptoms became very severe. I couldn't breathe well; I didn't sleep, and I was like "I am going to die here" . . . I said, "let me call a Boda [motorcycle taxi] to bring me so that I die [at] the hospital." It is from then that I decided to come to Nyakibale [Hospital].
>
> —37-year-old (yo) male patient

> I made the decision after she had developed severe diarrhea, vomiting, and wasting, so it is (at) that point that I decided to bring her here for care, otherwise she would have died by now. Because she could not even eat or turn in bed. . .we thought she was going to die.
>
> —60-yo seeking care for 4-month-old (mo) granddaughter

A substantial number of participants, approximately one-quarter, delayed going to the ED because of concern that they might have a "cultural illnesses" requiring treatment by a traditional practitioner or priest. Cultural illnesses are thought to be caused or exacerbated by factors allopathic medicine cannot treat. In these cases, the ED is utilized only when traditional treatment is thought to have been ineffective, or when traditional causes of the ailment have been sufficiently treated, enabling allopathic medicine to be effective. This approach was often advised by elders in the family or community.

There are [cultural illnesses] that attacked the patient, [which] came from my husband's family. We first thought that she was suffering from cultural illness. Therefore, we had to first try treating with traditional herbs and then [go] to the clinic.

—20-yo seeking care for her 11-mo daughter

The majority of study participants noted that they received advice before making the decision to seek EC. Most were advised to go directly to the ED, while some were advised to treat the underlying cultural disease before seeking allopathic care. Some participants noted that they did not tell ED staff about seeking traditional treatment for fear the staff might scold them for delaying seeking EC.

**Use of local facilities for stabilization and advice despite perception of inadequate services.** Over three-quarters of participants reported consulting a nearby allopathic or traditional medical care practitioner before presenting at the ED; approximately half described visiting multiple health care facilities and/or practitioners before ED presentation. Participants reported that obtaining care from multiple facilities is a common practice. Most providers utilized were accessible by walking or bicycling. These nearby practitioners were often consulted to assess the seriousness of the ailment and/or as an initial step in managing symptoms while gathering funds to cover the costs of higher-level care.

In some cases, participants chose to wait until this initial care was deemed ineffective before considering other options. However, most perceived the ED to offer superior care. Private clinics were reported by participants to lack necessary medication and materials, communicate poorly about the patient's condition and treatment, and generally provide inadequate care at high cost. Participants reported spending an average of $8 on care—a large sum in rural Uganda—before presenting at the ED.

It was lack of money, if you don't have money then you stay home or go to private clinics because you calculate the money for transport to getting here (the ED) . . .. then you decide to go to nearest clinic.

—64-yo women seeking care for herself

Some participants felt that private clinics postponed referral to capture revenue, even if they were unable to provide adequate care. Despite this, once a referral was made, participants often had to search for funds or gather supplies before seeking higher-level care.

. . . First, I sought health care in many private clinics. . .[Then], I decided to take the patient to [an outside] hospital and reaching there, they referred us here. . . I took the patient back home [for 2 days] so that I could prepare and come here. . .. I wanted to prepare what we were to use here [at the hospital], like food, and other things.

—20-yo seeking care for her 11-mo daughter

**Lack of resources to cover the direct, indirect, and opportunity costs of EC.** While most participants understood that payment was not required before ED care, they still felt they did not have the resources needed to seek EC. The majority of participants reported that lack of finances was the main reason for delay once danger was recognized. We queried perceptions regarding how much cash on hand participants felt they needed going to the ED. Speculations ranged broadly ($10-$150) and seemed unrelated to severity of patient complaint. Whatever the dollar amount mentioned, participants felt that receiving care at Karoli Lwanga Hospital would be "somewhat" to "extremely" unaffordable. In many cases it was not the direct cost of

the hospital that caused delays once a health problem was deemed severe, but indirect costs, including the expense of traveling to the hospital and paying for nonmedical supplies that would be needed by the patient and caregiver during their stay.

On average, patients traveled 12.6 km from their parish of residence to the ED (based on the distance from the center of the identified parish to the hospital determined through GIS measurement from a prior study [23]) at a mean cost of $6. Half reported that finding funds to pay for transportation was the greatest factor associated with delay. Once they obtained funding for transport, participants presented to the ED for evaluation, stabilization, and treatment and continued to search for funds to pay their hospital bill after obtaining EC.

> Right now, we are still soliciting for money. We thought about her life first, before thinking about money for medical care here. . . We were like "we will look for money later but let us first take her to the hospital."
>
> —25-yo seeking care for her 61-yo mother

The opportunity costs of health care seeking on work and ability to perform essential household activities impacted the decision to seek care. This especially impacted women who had to find alternative childcare or substitute agricultural labor in their absence from home. Extended family and neighbors were often asked to assist in supervising children, tending gardens and livestock, and ensuring the security of properties.

> "No one was left (at home). (The children) are at school, but on returning home they will prepare food for themselves. Their grandmother (lives) at a distance but sometimes she comes and helps. . .You can't leave the home alone. Also, farming being our business and it being a season of planting is hard."—28-yo seeking care for her 9-yo daughter

> I am the one responsible for my farms. I was harvesting millet and eventually got so sick I couldn't stand. Actually, this caused a long delay, for 4 days, because I was thinking about who will take care of my farms. I had even reached the point of refusing to be taken for medical care. I was worried about what I will feed my children after being discharged from the hospital, because I was leaving all my harvests in the farm. . .This disturbed [me] a lot.
>
> —41-yo male patient

**Inadequate transportation options, especially at night and in inclement weather.**
Transportation is a common source of delay for several reasons. Participants reported difficulty reaching a main road to access vehicles for hire. Once at the road, many reported difficulties locating a car or truck adequate for transporting patients with severe illness. Inclement weather, especially rain, further constrained participants' transport options. During monsoon season, motorcycle taxis—the most common form of transportation—were often unusable as dirt roads turned to mud following torrential rains.

> Nyakibale is very far away from us, otherwise I wouldn't seek medical care at [local] clinics. . .Poor roads and inadequate transport means delay us and make patients severe.
>
> —60-yo seeking care for her 4-mo granddaughter

Traveling at night was seen as an additional safety hazard. A third of participants reported waiting until morning to seek EC, despite a patient's worsening condition. Most patients who arrived at the ED at night did so by car, a far more expensive and difficult to arrange option.

The first barrier was that the patient got burnt at night and I couldn't get means of transport to bring us here quickly. Also, I was the only one at home with other young children, therefore I wouldn't get someone to watch the children under my care. Even on the following morning it rained heavily until 2:00pm, that's when I decided to come.

—40-yo women seeking care for her 4-yo daughter

## Discussion

Extensive literature is available on factors influencing illness-specific (eg, HIV, malaria) and demographic-specific (eg, pediatrics, maternity) health care seeking in Uganda [24–31]. However, only sparse research exists on challenges facing patients experiencing acute illness or injury, and little of this pertains to when formal EC is available in a rural area.

This study presents data on the first stage of multistage formative research centered on gaining insights into factors influencing treatment delay to EC by rural community members [32, 33]. A forthcoming study will explore possible interventions to increase timely EC access. This study identified four broad areas where interventions are warranted. It was informed by the Three Delays Model initially developed for obstetric emergencies [17] and applied more broadly to EC by Calvello et al [34] and Broccoli et al [35]. In the Three Delays Model [17]:

1. The first delay is the decision to seek care, including factors such as:

   - Availability of decision makers

   - Cultural perceptions that predispose community members to visit a traditional healer and/or self-treatment prior to seeking allopathic care

   - Distance to EC

   - Costs of accessing care

   - Availability of a caregiver to accompany the patient

   - Perceptions of the etiology and severity of an illness or injury, and the quality of care available

2. The second delay includes challenges in identifying and reaching a health facility, such as:

   - Distance to EC

   - Availability and cost of transport

   - Availability of a caregiver to accompany the patient

3. The third delay relates to receiving adequate and appropriate treatment at a health facility, including factors such as:

   - Appropriateness and quality of care offered

   - Resources at hand

   - Competent referral, as needed

We studied the first 2 delays in depth. Although participants alluded to the third delay, we did not explicitly include this delay in the study aims or directly investigate the quality of care or timeliness of referrals offered by other facilities. The interrelated nature of delays in the Three Delays Model and our own themes of delay are illustrated in Fig 1.

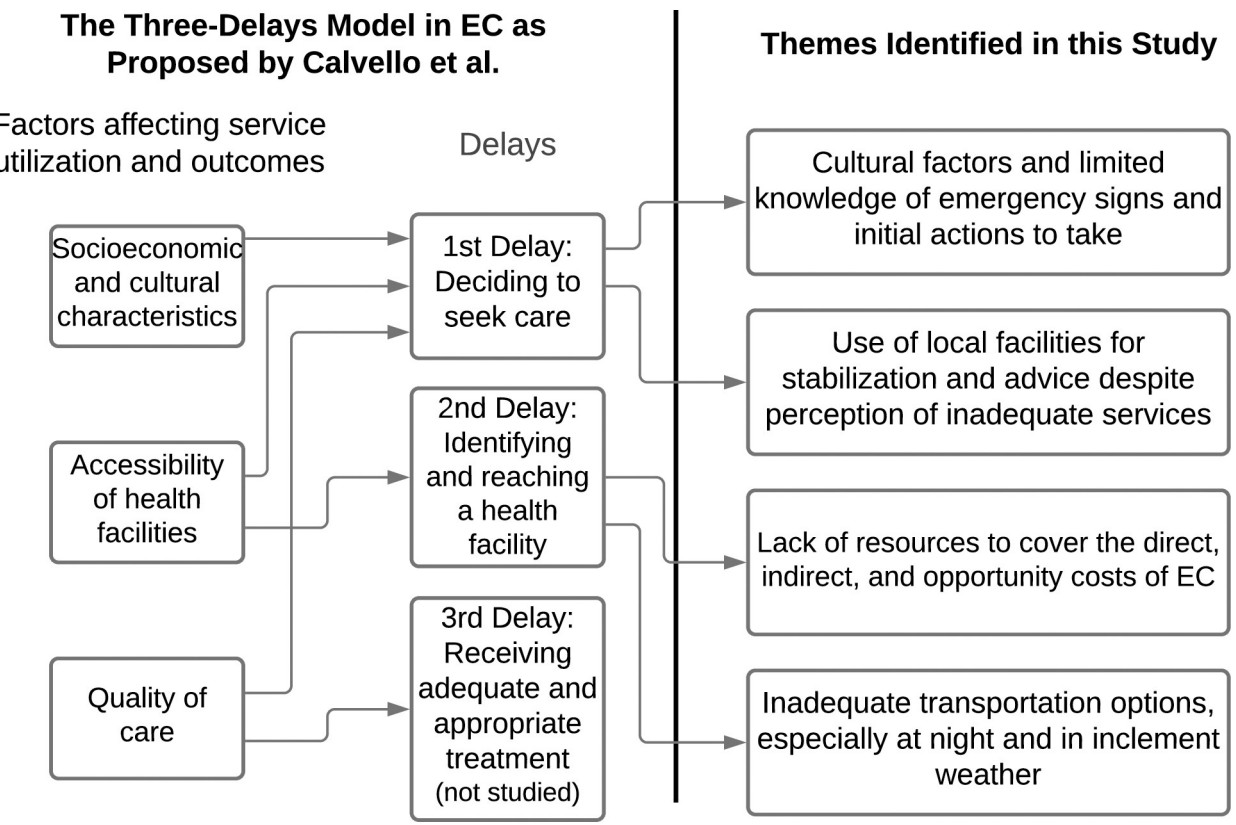

**Fig 1. Themes of delay of this study correlated with the three delays model.** Abbreviations: EC, emergency care.

Our results generally support the applicability of the three delays model to accessing EC at the community level. While we did not research delay in receiving adequate care at health facilities directly from interview participants' perceptions of delay in obtaining adequate treatment and/or referral at outlying facilities it can likely be implied that this delay is equally relevant, however further research is needed.

In this study, we found that, on average, 4 days elapsed between onset of the CC and the illness/injury being recognized as needing emergency treatment by the patient or caregiver. Another day passed between recognition of the need for EC and ED presentation.

### The first delay–Decision to seek care

**Cultural factors and limited knowledge of emergency signs and initial actions to take.** Delays often occurred in part because of a failure to recognize danger signs soon enough. Many participants traveled to the ED only when the patient's condition increased in severity to the point it was considered "life-threatening." Lack of knowledge or confidence regarding the danger signs heralding medical emergencies is consistent with earlier studies from rural areas of SSA [30, 35–37], as well as the findings of a recent systematic review of barriers to seeking EC in LMICs globally [38].

Home care (dietary change, bathing, bleeding control, and use of medicines purchased at informal pharmacies) is well documented in Ugandan studies of pediatric fever [27] and obstetric emergencies [37]. In this study, we found that once severe illness was recognized, few participants continued home care except as a stopgap measure while gathering resources for

consultation with an allopathic or traditional practitioner. In the case of injury, our findings are consistent with other East African studies reporting that few injury victims receive care at the scene of accidents or en route to a health facility, despite lengthy transports [39–41]. In keeping with prior studies, participants reported lack of first aid skills and supplies and expressed a desire for training [35, 36, 42, 43].

In the case of culturally salient folk illnesses, it is deemed pragmatic to seek traditional medicine before allopathic treatment. Traditional medicine is thought to improve efficacy of allopathic medicine, by removing cultural factors. Thus, not trying traditional medicine first might lead to a waste of resources, if only allopathic treatment is used and is ineffective. When the patient recovers after traditional treatment, before or in concert with allopathic treatment, dual causality of the illness is presumed [44, 45].

Based on this data we will conduct community based research which:

- Investigates how best to educate community members to recognize the danger signs of acute illness/injury and administer basic first aid.

- Assesses self-care practices.

- Seeks to better understand existing traditional medicine and explores the possibility of establishing collaborative relationships with traditional healers as a means of promoting rapid referral for EC.

**Use of local facilities for stabilization and advice despite perception of inadequate services.**   Before presenting to the ED, most study participants sought care from at least one local practitioner offering either traditional or allopathic treatment. Notably, we found that, participants did not choose to go to local allopathic clinics because they had confidence in, or good rapport with, local practitioners. Rather, they resorted to these clinics because of their accessibility and the perception that obtaining care at the ED would prove costly, even if funds were not demanded before treatment.

Similar to the findings of other studies conducted in Uganda, the general impression of participants was that government health stations and private clinics offer low-quality care and are best consulted as a stopgap measure until funds can be secured to travel to a hospital if the patient's condition does not improve [24, 27, 28, 30]. Interim treatment measures to allow time to monitor symptoms, identify possible etiologies, and select treatment options have also been documented in other settings [46].

Our study suggests that local clinics are resorted to for practical reasons, mainly as a stopgap measure while gathering funds needed for transportation. This suggests a need for both emergency transportation and referral systems in which local practitioners—both allopathic and traditional—feel they are members of a community of health care practice [32]. As members of such a community of practice, practitioners will need to receive training in danger signs warranting immediate referral to a EC.

### The second delay–Reaching care

**Lack of resources to cover the direct, indirect, and opportunity costs of EC.**   Cost is a major factor contributing to delays in seeking health care in rural Uganda and similar settings [30, 35, 36, 41, 42, 47, 48]. Three distinct types of cost need to be recognized when considering delays in obtaining EC: direct fees for health care services, indirect costs (eg, transport, food, lodging, and medical supplies/medications required for the accompanying caregiver), and opportunity costs of seeking care. Opportunity costs include both the financial impact of health care seeking on income-generating activities and loss of time to perform essential

household activities such as childcare [49]. Interviewees tended to focus on indirect and opportunity costs. The most prevalent indirect costs discussed were transport and the need for money to cover necessities like food while at the hospital. Caregivers repeatedly mentioned opportunity costs of accompanying patients to the hospital. Patient caregivers needed to find substitute labor to attend to essential household tasks in their absence. Concern about the status of one's children and farm were reported by nearly all participants interviewed. Asking kin or neighbors to watch children or farms has been noted to strain social relations, especially if assistance is needed during peak agricultural seasons [50].

Searching for a loan with little disposable income was a common source of delay and one that has been well described by several other studies in SSA [37, 51, 52]. While community saving and lending programs do exist in the Rukungiri District, they are not typically structured to provide emergency funds. Participants spoke of the difficulties entailed in borrowing money, especially between harvest seasons, as most are subsistence farmers. This elucidates the need for future formative research on implementation of EC specific loan schemes that build upon already-thriving rotating credit and microfinance programs within the community.

**Inadequate transportation options, especially at night and in inclement weather.** Difficulties obtaining transportation to the hospital was a theme in nearly all interviews. Interviewees often did not have cash on hand for transport, which is generally paid in advance and not available on credit. Once participants were finally able to gather funds, finding transport was often a challenge. The dangers of poorly maintained roads worsened the challenge, especially at night and during inclement weather. Many participants were left with no other option but to use less-safe forms of transport like motorcycle taxis. Multiple studies on health care seeking in Uganda and other countries in SSA have also identified transportation as a key cause of delay [35, 36, 39, 42, 47, 53, 54]. An unpublished study by the GEC Research Team conducted at this study site found that the distance from the patient home to the ED significantly impacted patient outcomes. In this study, compared to patients living < 5 km from the hospital, patients living 5–10 km away had 1.7 times the rate of mortality at 3 days, and those living >10 km away, 2.2 times [23].

This study indicates the need for an emergency transport system to be set up within the hospital catchment area. Following the period of this study, Nyakibale Hospital has collaborated with Humanitarian Aid for Uganda, a German nongovernmental organization, to set up an ambulance system in the district. Currently, 2 ambulances are staffed by volunteer ECPs and midwifes who carry an emergency phone. Their phone number has been widely distributed in the community. To facilitate timely ED access, payment for ambulance service is not required at the time of transport. Given that mobile phone service is rapidly increasing in rural Uganda, future formative research will investigate whether arrangements might be made with vehicles for hire to pick up patients for a set fee, if and when an ambulance service cannot be accessed.

**The third delay—Health care services.** Referral delay from local healthcare facilities was beyond the scope of this study and will require a study of practitioners' recognition of danger signs, the quality of care available at local clinics, and financial motivations for delaying a patient before referral. The collection of such data in the future is needed to inform health service interventions which provide bidirectional rewards for both local practitioners and the ED while promoting timely EC referral; in order to be successful, local practitioners will need to gain status by facilitating timely referrals.

## Limitations

Four limitations of this study may be noted. First, interviews were held at the ED. While this allowed access to participants currently utilizing EC and likely reduced recall bias through

timely data collection, participants may have been uncomfortable sharing negative perceptions of ED services while receiving treatment. Second, the study did not include patients who were unable to present at the ED and therefore may not have identified additional barriers to EC access and utilization by the most vulnerable people in the hospital's catchment area. We plan to address this limitation in forthcoming research utilizing community-based focus groups. Third, we did not study delayed referral to ED from local health facilities, nor the quality of care offered at these facilities. Forth, this study was not designed to quantitatively assess household level factors potentially influencing delay in presentation to the ED, such as, age, gender, birth order, and household structure.

A final limitation of this study relates to sampling and the generalizability of the data collected. The objective of this qualitative study was to identify the range of factors that affect care seeking behavior at a relatively newly established ED. A purposeful sample was employed toward this end. As such, this data is not generalizable to other EDs although it is reasonable to hypothesize that similar factors leading to treatment delay may apply.

## Conclusions

EC can substantially reduce preventable complications and mortality in LMICs when the signs of acute illness and injury are recognized early, referral networks are established, and community members view EDs as accessible and affordable. Setting up EDs and training staff in LMICs is only the first step in ensuring that availability of EC services. In Uganda, a well-functioning ED was established, and a health care team trained; however, patient presentations to the ED are often delayed, reducing its potential benefits. This qualitative research study identified major factors contributing to delays in visiting the ED. It corroborates and builds on the findings of other non-EC related health service delivery studies utilizing the Three Delays Model and recommends the need for the next phase of formative research to identify and weigh the advantages and disadvantages of various community-based EC educational, financial, transport, and referral network interventions.

## Supporting information

**S1 File. Interview guide.**
(PDF)

**S2 File. Code book with supplemental quotes.**
(PDF)

**S3 File. ED patients' chief complaints.**
(PDF)

## Acknowledgments

The authors sincerely thank the Global Emergency Care Collaborative Investigators Mark Bisanzo, Heather Hammerstedt, Stacey Chamberlain and Bradley Dreifuss for their work in developing the EC and research infrastructure that made this study possible. Additionally, we would like to thank the GEC Research Team Members Charles Ndyamwijuka, Adrine Kusasira and Nelly Mbabazi for their tireless work on the research tool, interviews and data analysis, Hilary Kizza, GEC Program Coordinator, for his administrative support, Joseph Kalanzi and Edgar Mugema Mulogo for their assistance conceptualizing this project in the Ugandan context, and Deborah Stein for providing language editing of the manuscript. The Hospital Management Team at Karoli Lwanga Hospital also provided invaluable support for this study and

the ECP program. Finally, we would like to thank the Emergency Care Practitioners for their assistance with patient recruitment and data analysis, as well the critical EC they provide

## Author Contributions

**Conceptualization:** Ashley E. Pickering, Heather M. Dreifuss, Mark Nichter, Bradley A. Dreifuss.

**Data curation:** Ashley E. Pickering, Heather M. Dreifuss, Charles Ndyamwijuka, Bradley A. Dreifuss.

**Formal analysis:** Ashley E. Pickering, Heather M. Dreifuss, Charles Ndyamwijuka, Mark Nichter, Bradley A. Dreifuss.

**Funding acquisition:** Bradley A. Dreifuss.

**Investigation:** Ashley E. Pickering, Heather M. Dreifuss, Charles Ndyamwijuka, Mark Nichter, Bradley A. Dreifuss.

**Methodology:** Ashley E. Pickering, Heather M. Dreifuss, Mark Nichter, Bradley A. Dreifuss.

**Project administration:** Ashley E. Pickering, Bradley A. Dreifuss.

**Supervision:** Bradley A. Dreifuss.

**Validation:** Heather M. Dreifuss, Charles Ndyamwijuka, Bradley A. Dreifuss.

**Visualization:** Heather M. Dreifuss, Bradley A. Dreifuss.

**Writing – original draft:** Ashley E. Pickering, Heather M. Dreifuss, Charles Ndyamwijuka, Mark Nichter, Bradley A. Dreifuss.

**Writing – review & editing:** Ashley E. Pickering, Heather M. Dreifuss, Charles Ndyamwijuka, Mark Nichter, Bradley A. Dreifuss.

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
