## [Decision Letter · Decision Letter 0]

6 Aug 2021

PONE-D-21-13731

Building It so They Will Come: Interviews With Emergency Department Patients and Their Caregivers on the Challenges to Emergency Care Utilization in Rural Uganda - a Grounded Theory Approach

PLOS ONE

Dear Dr. Pickering,

Thank you for submitting your manuscript to PLOS ONE. After careful consideration, we feel that it has merit but does not fully meet PLOS ONE’s publication criteria as it currently stands. Therefore, we invite you to submit a revised version of the manuscript that addresses the points raised during the review process.

Congratulations on a very significant and impactful project conducted with sound qualitative science. The utilization of frameworks and rigorous methods are excellent, the but integration of these frameworks throughout the manuscript more fundamentally and streamlining the discussion to interpretation of the results and support of next steps rather than reiteration of the results are needed. Please mention and utilize a reporting guideline in your methods. These changes while listed as major given the foundational aspect of the framework integration, can also be viewed as minor as they are less about the methods/analysis and more about the description of the results and interpretation. Please also see the comments from Reviewer #1 about data sharing.

We look forward to receiving your revised manuscript.

Kind regards,

Catherine A Staton, M.D., MSc

Academic Editor

PLOS ONE

Journal Requirements:

Reviewers' comments:

Reviewer's Responses to Questions

**Comments to the Author**

1. Is the manuscript technically sound, and do the data support the conclusions?

Reviewer #1: Yes

Reviewer #2: Yes

Reviewer #3: Yes

2. Has the statistical analysis been performed appropriately and rigorously? 

Reviewer #1: Yes

Reviewer #2: N/A

Reviewer #3: I Don't Know

3. Have the authors made all data underlying the findings in their manuscript fully available?

Reviewer #1: No

Reviewer #2: Yes

Reviewer #3: Yes

4. Is the manuscript presented in an intelligible fashion and written in standard English?

Reviewer #1: Yes

Reviewer #2: Yes

Reviewer #3: Yes

5. Review Comments to the Author

Reviewer #1: 1.Is the manuscript technically sound, and do the data support the conclusions?

Overall a well written manuscript on an important topic. I would suggest considering adding additional quotes to support your findings.

2. Has the statistical analysis been performed appropriately and rigorously?

Methodologically sound. Mention in your methods what reporting guideline was used for this manuscript.

3. Have the authors made all data underlying the findings in their manuscript fully available?

You state that the data is available in the supporting files, however this is only the codebook and not the transcripts. If the IRB approves releasing the full transcripts, that would qualify as all data is fully available without restriction.

If not, explain the reasoning for restricted/no access, and consider adding quotes to the full codebook in the supporting information.

Overall comments:

Interesting and important study, with robust qualitative methods, and integrating ongoing local efforts to increase access to EC.

Discussion could benefit from some minor restructuring. Lots of re-iteration of the results. The three delay framework gets introduced, but throughout the discussion it is not clear how it was applied. Consider adding the headings from the identified delays you are discussing, and how those could be addressed (the 4 main points for action you mention in the Conclusion)

It is not clear to me from this paper why community based interventions should be the next focus, instead of provider centered interventions and Health systems strategies.

The Corresponding and first author is affiliated with GEC, the setting the interviews took place and patients were treated, and was present during the interviews. How was a potential power dynamic between the interviewer team and patients managed, so that participants did not feel pressured to participate?

Feedback by line:

Line 36: How was data saturation determined?

Line 40-42: Does this include patient and caregivers? The remainder of the participant characteristics is centered around the patients. It would be helpful to separate gender and occupation for patient and caregivers

Line 40: How have you identified that interventions in the community are necessary instead of possible HS based interventions/ provider interventions etc.

Line 66: remove brackets,

Line 75-83: Good summary of the need for action/ research

103-107: references available?

96-123: What information is essential for the reader to understand your research? Consider shortening this section

Line 139: Why >12 hours after presenting to the ED?

Line 141: How was determined who has significant experience with challenges in EC access and utilization?

Line 154: Was assent obtained from the patients, if the caregivers were interviewed on their behalf, since the interview questions were directly related to the patient experience?

164: add the reference to the three delays model

166/67: some additional detail on the translation process and validation. How was consensus reached?

179: Why are patients utilizing the ED at the time of interview not appropriate for member checking? Please clarify

186: growing field of research now. Consider clarifying, that in 2017, little was known about delays in accessing care

195: refer to the supporting information or where the codebook can be found

203: Are some caregivers of patients also included in the study? Highlight the most important characteristics in text instead of just referring to the table.

206: Table 1: Not immediately clear what the participant sample looks like. Consider splitting the table up in patient and caregiver characteristics

207: Table 2: What are other medical complaints? (Given that it is 26% of all medical cases.

234: remove “a”

260: is that a quote? “Failed” How were failed treatments perceived by the participants?

264: if 8$ is a large sum, provide some context, eg average net income

286: was this data provided by the participants during the interview? Consider clarifying that this is based on their recall and no additional information sources.

295: What is meant with opportunity costs?

344: Was the interview guide developed based on the three delay model or applied afterwards?

347: Figure 1: make sure to demonstrate which your own themes of delay are based on the data.

Reviewer #2: The authors present an important manuscript qualitatively evaluating reasons for delays in seeking emergency care in rural Uganda as elicited from patients and caregivers. A sound framework and underlying theoretical model were used.

- 50 surveys for a qualitative project is a large number, and data saturation was already reached at 25. Were there any additional themes identified in the last 25?

- Table 2 separates CC by medical and injury, yet in table 3 and elsewhere, CC is separated by medical illness, injury, and chronic disease. This should be consistent across tables.

- Further clarification of chief complaint is needed. The authors indicate that only acute presentation of chronic conditions were included in the sample, however, in Table 3 there are chief complaints that have a duration of 3 years noted. This does not make sense for an acute presentation of an illness or a medical illness.

- Given the distinct differences in trauma and medical emergency complaints, it would be interesting to note if there were any differences in subthemes between these 2 groups. For example, "cultural illnesses" requiring treatment by a traditional healer, seem to be more for primarily medical emergency conditions.

- Limitations section notes subjectivity of the data on the quality of local clinics. This is not a limitation and should be removed. Given that this is qualitative research, these are the noted perceptions of the interviewees and important as such in terms of understanding behaviors behind using or not using these services.

Other errors:

Line 164: need to cite The Three Delays model

Line 265: missing a period after ED

Line 351: missing the word "needing" before emergency treatment?

Reviewer #3: The goal of the study was to assess/explore the barriers for accessing and utilizing emergency care in rural Uganda. Which I think the study achieved. It was done by using semistructured interviews. 50 patients and caregivers were interviewed. Good description of method with construction of the interview questions, how the authors tried to avoid bias, ethical approvoals etc. Four major themes of delay appeared: 1) "Limited knowledge of emergency signs and of initial actions to take"; 2) "Use of local facilities for stabilization and advice despite perception of in adequate services"; 3) "Lack of resources to cover the direct, indirect and opportunity costs of EC"; 4) "In adequate transportation options, especially at night and inclement weather". In the discussion the theory of the Three Delays model (where focus was on the two first ones) was discussed and applied to the results and where a correlation was found which I find was good. Four issues (1) teaching programs, 2) EC loans, 3) emergency transport and communicating systems, 4) referral system) are pointed out in the conclusion that needs immediate attention which are all good and in line with the findings but I would have liked them to have been motivated a little bit more and maybe already pointed out in the discussion section.

6. PLOS authors have the option to publish the peer review history of their article (what does this mean?). If published, this will include your full peer review and any attached files.

Reviewer #1: No

Reviewer #2: No

Reviewer #3: No

---

## [Author Response · Author response to Decision Letter 0]

22 Sep 2021

Catherine A Staton, M.D., MSc

Academic Editor

PLOS ONE

September 20, 2021 

Dear Dr Staton,

Thank you for your in-depth review of our work. We are thrilled that the reviewers find this work to be as important as we do. The insights and reviewer comments helped to clarify and strengthen the paper. 

Overall, reviewer comments helped us to clarity the scope of our work both internally and for the reader. The aim of this study was increasing understanding of the challenges and delays to EC access in the community. While we are aware that there is significant work to be done to facilitate timely, quality emergency care at the health facility level this was not the focus of our aims or methods. Ultimately, this study is an initial formative work seeking to understand the emergency care seeking behavior of community members utilizing the established ED. As such the focus was on understanding challenges to emergency care access and not on specific interventions to address these challenges which would be acceptable to and effective for the community. While we make several general recommendations for areas of focus for ongoing research on interventions, this is not the focus of our paper. Rather, as this one of the first studies to describe challenges to seeking care at an established emergency department we contextualize our results within available literature on care seeking in acute illness and injury, drawn largely from the pediatrics and obstetrics literature. We believe that this increases the value of our results in adding to understanding of emergency care seeking behavior and in providing a basis for ongoing research. We have clarified our aims within the discussion section and edited our title to help the reader better understand the scope of the work. We integrated the three delays model into the discussion section and added additional discussion around the support our results lend to the model’s applicability to emergency care access. 

We utilized the COREQ guideline in conducting our study and drafting our paper to ensure compressive reporting. This has been added to the methods section.

Our IRB does not allow for data sharing, so we will be unable to publish the interview transcripts. However, as suggested we have added a supplement including our code book with further representative quotes to ensure readers have as broad an access as ethically possible.

We have addressed all reviewer comments inline below.

We look forward to your review of our revisions and final decision regarding publication. 

Thank you,

Ashley Pickering, MD, MPH

Corresponding Author

University of Maryland Medical Center

Department of Emergency Medicine

110 S. Paca St, Suite 600

Baltimore, MD 21201

Department Phone: 667-214-2181 

Reviewer comments addressed in-line (line numbers refer to markup):

Reviewer #1: 1.Is the manuscript technically sound, and do the data support the conclusions?

Overall a well written manuscript on an important topic. I would suggest considering adding additional quotes to support your findings.

Several additional quotes were added to the results section. We also added supplemental information containing our codebook with many additional quotes by subtheme.

2. Has the statistical analysis been performed appropriately and rigorously?

Methodologically sound. Mention in your methods what reporting guideline was used for this manuscript.

Added line 286/287 - The Consolidated Criteria for Reporting Qualitative Research (COREQ) was utilized to ensure comprehensive reporting.

3. Have the authors made all data underlying the findings in their manuscript fully available?

You state that the data is available in the supporting files, however this is only the codebook and not the transcripts. If the IRB approves releasing the full transcripts, that would qualify as all data is fully available without restriction.

If not, explain the reasoning for restricted/no access, and consider adding quotes to the full codebook in the supporting information.

As above - Our IRB does not allow for data sharing, so we will be unable to publish the interview transcripts. However, as suggested we have added a supplement including our code book with further representative quotes. 

Overall comments:

Interesting and important study, with robust qualitative methods, and integrating ongoing local efforts to increase access to EC.

Discussion could benefit from some minor restructuring. Lots of re-iteration of the results. The three delay framework gets introduced, but throughout the discussion it is not clear how it was applied. Consider adding the headings from the identified delays you are discussing, and how those could be addressed (the 4 main points for action you mention in the Conclusion)

These changes were made in the discussion section. The Three Delays model was applied as a framework for structuring discussion and contextualization of key results (as is common in most discussion sections) within a review of the limited available literature and ideas of how these could be addressed were moved from the conclusions to the discussion.

It is not clear to me from this paper why community based interventions should be the next focus, instead of provider centered interventions and Health systems strategies.

As above - The aim of this study was increasing understanding of the challenges and delays to EC access in the community. While we are aware that there is significant work to be done to facilitate timely, quality emergency care at the health facility level this was not the focus of our aims or methods.

The Corresponding and first author is affiliated with GEC, the setting the interviews took place and patients were treated, and was present during the interviews. How was a potential power dynamic between the interviewer team and patients managed, so that participants did not feel pressured to participate?

Added line 242-244 - AP was not involved in providing clinical care. NC approached patients to gauge interest in study participation, limiting pressure to participate.

Feedback by line:

Line 36: How was data saturation determined?

This statement as deleted as the abstract is not the appropriate word count for an in-depth discussion of data saturation. 

Question is addressed 282-283 - Data saturation was determined by additional interviews failing to demonstrate additional themes of delay.

Line 40-42: Does this include patient and caregivers? The remainder of the participant characteristics is centered around the patients. It would be helpful to separate gender and occupation for patient and caregivers

Addressed 65-67: The 50 ED patients for whom care was sought (mean age 33) with approximately even distribution of gender, as well as occupation (none, subsistence farmers and business owner).

Line 40: How have you identified that interventions in the community are necessary instead of possible HS based interventions/ provider interventions etc.

Addressed line 90 – 92: Interventions are warranted to address each of the four major reasons for treatment delay. The next stage of formative research will generate intervention strategies and assess the opportunities and challenges to implementation with community and health system stakeholders

Line 66: remove brackets,

Removed

Line 75-83: Good summary of the need for action/ research

Thank you

103-107: references available?

Reference added

96-123: What information is essential for the reader to understand your research? Consider shortening this section

Removed non-essential data 

Line 139: Why >12 hours after presenting to the ED?

≥12 hours was set as a benchmark based on advice from the Ugandan clinicians working in the ED 

Line 141: How was determined who has significant experience with challenges in EC access and utilization?

Addressed lines 205/206 - based on conversations with ED staff

Line 154: Was assent obtained from the patients, if the caregivers were interviewed on their behalf, since the interview questions were directly related to the patient experience?

Addressed lines 220-221: In the case of caregiver interviews, assent was obtained from patients old enough, or alert enough to provide it. In most instances this applied to the young children, the elderly and those with severe illness.

164: add the reference to the three delays model

Reference added

166/67: some additional detail on the translation process and validation. How was consensus reached?

Addressed lines 250-252 - If there was a discrepancy in translation/interpretation the audio recording was reviewed by NC and KA together and discussed until consensus was reached. 

179: Why are patients utilizing the ED at the time of interview not appropriate for member checking? Please clarify

Addressed lines 254-255: Due to the nature of the sample — patients utilizing the ED — typically participants were not available on site when transcription and translation were complete. The IRB limited collection of contact information to ensure privacy, therefore transcripts were not returned to interviewees for review.

186: growing field of research now. Consider clarifying, that in 2017, little was known about delays in accessing care

Added this clarification on line 261-262

195: refer to the supporting information or where the codebook can be found

Codebook with additional quotes included as Appendix S2

203: Are some caregivers of patients also included in the study? Highlight the most important characteristics in text instead of just referring to the table.

Addressed lines 290-298- The 50 ED patients for whom care was sought (13 of whom were interviewed directly) had a mean age of 33 years. Patients had an approximately even distribution of gender, as well as occupation (none, subsistence farmers and small business owner). Demographics and CCs for the ED patients are illustrated in Table 1. A full list of CCs is included as Appendix S3. Fifty interviews were completed, 37 with caregivers of ED patients and 13 with the ED patients themselves. Interviewees had a mean age of 38 years, and the majority were women, worked as subsistence farmers, and had received primary education. The demographics of interview participants are presented in Table 2. 

206: Table 1: Not immediately clear what the participant sample looks like. Consider splitting the table up in patient and caregiver characteristics

Clarified relationship of interview participant to patient. Switched the ordered or Tables 1 and 2 to help clarify the demographics for all patients, prior to discussing the overlap between patients and caregivers.

207: Table 2: What are other medical complaints? (Given that it is 26% of all medical cases.

Added all list of all chief complaints in Appendix S3

234: remove “a”

Removed 

260: is that a quote? “Failed” How were failed treatments perceived by the participants?

Addressed line 274 - treatment is thought to have been ineffective

264: if 8$ is a large sum, provide some context, eg average net income

As address in the introduction, subsection The Rukungiri District, 78% of the population are subsistence farmers. We have not been able to identify any source of average net income for the district, the western region, or even rural areas of Uganda. Overall average income for Ugandan is significantly higher than in the Rukungiri District due to the impact of urban areas with increased individuals with formal employment.

286: was this data provided by the participants during the interview? Consider clarifying that this is based on their recall and no additional information sources.

Clarification added lines 444-448 - based on the distance from the center of the identified parish to the hospital determined through GIS measurement for a prior study (23)

295: What is meant with opportunity costs?

Clarification added lines 457-458– The opportunity costs of health care seeking on work and ability to perform essential household activities impacted the decision to seek care

344: Was the interview guide developed based on the three delay model or applied afterwards?

Addressed line 510-511 - The interview guide and a framework for the themes we identified were informed by the Three Delays Model

347: Figure 1: make sure to demonstrate which your own themes of delay are based on the data.

Figure updated to clarify

Reviewer #2: The authors present an important manuscript qualitatively evaluating reasons for delays in seeking emergency care in rural Uganda as elicited from patients and caregivers. A sound framework and underlying theoretical model were used.

- 50 surveys for a qualitative project is a large number, and data saturation was already reached at 25. Were there any additional themes identified in the last 25?

Addressed lines 270-282: These themes and subthemes constituted a codebook, included as Appendix S2, facilitating coding of an additional 25 interviews completed to capture a broader sample of demographic groups and chief complaints across both dry and rainy seasons, given differences in EC accessibility and subsistence agriculture activities. Data saturation was determined when interviews failed to demonstrate additional themes related to treatment delay.

- Table 2 separates CC by medical and injury, yet in table 3 and elsewhere, CC is separated by medical illness, injury, and chronic disease. This should be consistent across tables.

We separated CC by medical and injury. History of chronic disease was found to significantly increase delay for acute illness or injury, regardless of if it was related to the chronic disease, so this data as pulled on in table 3 to illustrate this finding.

- Further clarification of chief complaint is needed. The authors indicate that only acute presentation of chronic conditions were included in the sample, however, in Table 3 there are chief complaints that have a duration of 3 years noted. This does not make sense for an acute presentation of an illness or a medical illness.

Addressed lines 314-320 - The most notable trend related to treatment delay at the ED was history of chronic disease. ED patients with acute presentations and history of underlying chronic conditions experienced substantially longer delays in reaching the ED than otherwise healthy patients. Among chronically ill patients, recognition of acute symptoms needing EC ranged from 12 hours – 14 days prior to arriving at the ED, with an average of 6 days. For all patients included in the study the mean duration from recognition of symptoms necessitating EC to presentation at the ED as 1 day. Patients with chronic disease and their caregivers reported challenges to decision-making including perceptions that:

- Given the distinct differences in trauma and medical emergency complaints, it would be interesting to note if there were any differences in subthemes between these 2 groups. For example, "cultural illnesses" requiring treatment by a traditional healer, seem to be more for primarily medical emergency conditions.

Any differences identified have been highlighted in the paper

- Limitations section notes subjectivity of the data on the quality of local clinics. This is not a limitation and should be removed. Given that this is qualitative research, these are the noted perceptions of the interviewees and important as such in terms of understanding behaviors behind using or not using these services.

This was removed from the limitations and is addressed in the discussion section.

Other errors:

Line 164: need to cite The Three Delays model

Citation added.

Line 265: missing a period after ED

Addressed.

Line 351: missing the word "needing" before emergency treatment?

Addressed.

Reviewer #3: The goal of the study was to assess/explore the barriers for accessing and utilizing emergency care in rural Uganda. Which I think the study achieved. It was done by using semistructured interviews. 50 patients and caregivers were interviewed. Good description of method with construction of the interview questions, how the authors tried to avoid bias, ethical approvoals etc. Four major themes of delay appeared: 1) "Limited knowledge of emergency signs and of initial actions to take"; 2) "Use of local facilities for stabilization and advice despite perception of in adequate services"; 3) "Lack of resources to cover the direct, indirect and opportunity costs of EC"; 4) "In adequate transportation options, especially at night and inclement weather". In the discussion the theory of the Three Delays model (where focus was on the two first ones) was discussed and applied to the results and where a correlation was found which I find was good. Four issues (1) teaching programs, 2) EC loans, 3) emergency transport and communicating systems, 4) referral system) are pointed out in the conclusion that needs immediate attention which are all good and in line with the findings but I would have liked them to have been motivated a little bit more and maybe already pointed out in the discussion section.

The four broad intervention strategies identified by the reviewer were moved to the discussion section. We provide limited discussion of interventions as this is largely outside the scope of this work and requires further research is addressed in the letter above.

---

## [Decision Letter · Decision Letter 1]

18 Feb 2022

PONE-D-21-13731R1Getting to the Emergency Department in Time: Interviews With Patients and Their Caregivers on the Challenges to Emergency Care Utilization in Rural Uganda - a Grounded Theory ApproachPLOS ONE

Dear Dr. Pickering,

Thank you for submitting your manuscript to PLOS ONE. After careful consideration, we feel that it has merit but does not fully meet PLOS ONE’s publication criteria as it currently stands. Therefore, we invite you to submit a revised version of the manuscript that addresses the points raised during the review process.

The revied manuscript has been returned to the reviewers who have expressed overall positive feedback following the revision of your mansucript.

However reviewer 1 has provided additional comments for further clarification which we believe can further strength the mansucript.

Could you please carefully revise the manuscript to address all comments raised?

We look forward to receiving your revised manuscript.

Kind regards,

Lucinda Shen, MSc

Staff Editor

PLOS ONE

Journal Requirements:

Reviewers' comments:

Reviewer's Responses to Questions

**Comments to the Author**

1. If the authors have adequately addressed your comments raised in a previous round of review and you feel that this manuscript is now acceptable for publication, you may indicate that here to bypass the “Comments to the Author” section, enter your conflict of interest statement in the “Confidential to Editor” section, and submit your "Accept" recommendation.

Reviewer #1: All comments have been addressed

Reviewer #2: All comments have been addressed

Reviewer #3: All comments have been addressed

2. Is the manuscript technically sound, and do the data support the conclusions?

Reviewer #1: Partly

Reviewer #2: Yes

Reviewer #3: Yes

3. Has the statistical analysis been performed appropriately and rigorously? 

Reviewer #1: Yes

Reviewer #2: Yes

Reviewer #3: Yes

4. Have the authors made all data underlying the findings in their manuscript fully available?

Reviewer #1: No

Reviewer #2: Yes

Reviewer #3: Yes

5. Is the manuscript presented in an intelligible fashion and written in standard English?

Reviewer #1: Yes

Reviewer #2: Yes

Reviewer #3: Yes

6. Review Comments to the Author

Reviewer #1: Thank you for addressing our comments to the previous version of the manuscript. I enjoyed reading the article and think you have made the aims of this project and placing it into the larger research focus of your team much clearer. The integration of the three-delay framework and discussion especially are very well written.

I think the abstract is currently not reflecting the changes you made to the manuscript appropriately. Additionally, I suggest some restructuring and clarifications in the methods section, to further increase the rigor of the study. Please see more specific suggestions below.

Line5-7: In the response letter to the editor, you very clearly state the aims of this study. I suggest using that in the abstract. “Implementation of effective EC requires assessment of socioeconomic, cultural, and structural factors leading to treatment delay.” Might lead the reader to believe you will be focusing on the implementation of strategies, instead of first understanding the challenges.

Line 13-14: not clear how translating and transcribing the interviews allow for a grounded theory approach and thematic analysis. I’d suggest simply stating what analysis approach you chose. If there is space, I suggest stating why you chose 50 participants as a sample size or if it was driven by saturation.

18: I know space is very limited and you address it later in text, but some readers might wonder why there are almost three times as many caregivers interviewed as patients? Were they used as a proxy because the patients were not available? Could be as simple as stating “patients or caregivers of patients being unavailable were interviewed on their behalf.”

16-19: you addressed the age questions in the revisions, however I still find it slightly confusing. The mean ages are not clear. All patients had an average age of 33, but all interviewees were 38 years on average? This sentence seems out of place, as you switch back to the patients’ experiences afterwards. I suggest removing that sentence altogether

Line 58-60. Much clearer description of the aims, using a more qualitative language than in the abstract

Line 61-108: appreciate the explanations and giving context to the reader. Well written and informative.

Methods:

- I suggest using sub-headings for the methods section to make it more reader friendly.

- Provide a brief statement at the beginning of the methods that outlines the study design and analysis approach. This will be especially beneficial for readers that are not well versed in qualitative research designs.

- You state in the title that a grounded theory approach was used, however that is a very specific iterative approach, that requires more details on how the approach was followed, vs an analysis based in phenomenology for example. Make sure you truly used a complete grounded theory approach, rather than utilizing certain elements based in grounded theory. Specify how the Framework Method helps to follow a grounded theory approach.

Line 111: you justify the choice for the 12 hour time mark in your response letter, I suggest adding that here. It provides additional rigor to the methods.

Line 114: information rich cases. Briefly state how you determined who’d have significant experience with challenges in EC access prior to having interviewed them. What were the characteristics the ED staff suggested you to include to meet this criteria?

Line 129-131: I am curious if severe illness of the patient was a common reason for caregivers to refuse the interviews. I imagine this could be a very stressful and potentially challenging interview for caregivers and patients to have. Where any resources put in place, in case they needed support?

Line 132-145: good description of the interview guide development. Was piloting of the questions performed? If not, what was the rationale?

Line 146: briefly describe how the training in qualitative interview-based methods looked like. Didactics only, practice sessions, etc.

Lines 172- 183: the data saturation statement is still not clear. I understand the rationale about adding more interviews to have a broader demographic profile, but it still sounds like saturation was reached after 25 interviews and not after 50. How exactly was saturation determined? Provide more details on the type of saturation (thematic vs data saturation)

Line 194: I appreciate switching the Tables 1 and 2. Much clearer

198: the distinction of “delay in ED presentation” and “Themes in Delay” is not very clear. Delay in ED provides the quantitative explanation of delays, whereas themes in delay further explains and seeks to understands the delays.

220: A brief overview of themes and subthemes in table form or some visual could be helpful

239: Could you include the specific word used for “cultural illness” in the local dialects?

259: Do you have information on what the maximum number of facilities visited before the ED presentation was?

288: I understand you mentioned in your response letter that it is difficult to provide context on how much a dollar sum is in the Ugandan context, specifically in this region, where a national net income would likely be an overestimate. But I think it is important to emphasize somehow, that 150$ is a significant sum and potentially catastrophic healthcare expenditure. Maybe you could add some small section in the description of the setting (lines 68-69)

324: change to past tense, check for consistency throughout

343: Discussion makes the objectives and planned next steps much clearer, positioning this paper in the ongoing research efforts. Good integration of the available literature.

Reviewer #2: (No Response)

Reviewer #3: I think you have addressed and covered the reasons for delay in careseeking of this area in Uganda well and got results that are in line with the Three Delays Model. These results are discussed thoroughly in the discussion as well as suggestions and thoughts on how to move forward are pointed out. With all studies there are limitations and I think you are aware of them and address them wisely.

7. PLOS authors have the option to publish the peer review history of their article (what does this mean?). If published, this will include your full peer review and any attached files.

Reviewer #1: No

Reviewer #2: No

Reviewer #3: No

---

## [Author Response · Author response to Decision Letter 1]

3 May 2022

Reviewer Comments to the Author

Reviewer #1: Thank you for addressing our comments to the previous version of the manuscript. I enjoyed reading the article and think you have made the aims of this project and placing it into the larger research focus of your team much clearer. The integration of the three-delay framework and discussion especially are very well written.

I think the abstract is currently not reflecting the changes you made to the manuscript appropriately. Additionally, I suggest some restructuring and clarifications in the methods section, to further increase the rigor of the study. Please see more specific suggestions below.

Line5-7: In the response letter to the editor, you very clearly state the aims of this study. I suggest using that in the abstract. “Implementation of effective EC requires assessment of socioeconomic, cultural, and structural factors leading to treatment delay.” Might lead the reader to believe you will be focusing on the implementation of strategies, instead of first understanding the challenges.

Line 6-7 changed to: “This study seeks to understand the emergency care seeking behavior of community members utilizing the established ED.” 

Line 13-14: not clear how translating and transcribing the interviews allow for a grounded theory approach and thematic analysis. I’d suggest simply stating what analysis approach you chose. If there is space, I suggest stating why you chose 50 participants as a sample size or if it was driven by saturation.

Addressed on lines 11-14: “Semistructured interviews addressing actions taken before seeking EC and delays to presentation once the need for EC was recognized were conducted until theoretical saturation and a diverse sample were obtained. An interdisciplinary and multicultural research team conducted thematic analysis based on descriptive phenomenology.”

18: I know space is very limited and you address it later in text, but some readers might wonder why there are almost three times as many caregivers interviewed as patients? Were they used as a proxy because the patients were not available? Could be as simple as stating “patients or caregivers of patients being unavailable were interviewed on their behalf.”

Addressed lines 18-19: “Interviews were conducted with 13 ED patients and 37 caregivers, on the behalf of patients when unavailable.”

16-19: you addressed the age questions in the revisions, however I still find it slightly confusing. The mean ages are not clear. All patients had an average age of 33, but all interviewees were 38 years on average? This sentence seems out of place, as you switch back to the patients’ experiences afterwards. I suggest removing that sentence altogether.

This was removed and helped to make space for other clarifications.

Line 58-60. Much clearer description of the aims, using a more qualitative language than in the abstract

Thank you. 

Line 61-108: appreciate the explanations and giving context to the reader. Well written and informative.

Thank you. 

Methods:

- I suggest using sub-headings for the methods section to make it more reader friendly.

These have been added. 

- Provide a brief statement at the beginning of the methods that outlines the study design and analysis approach. This will be especially beneficial for readers that are not well versed in qualitative research designs.

This was added on lines 122-128: “To better understand the challenges rural Ugandans face in accessing and utilizing EC we purposefully sampled and interviewed patients presenting to the ED more than 12 hours after onset of chief complaint or their caregiver when the patient themselves was unavailable. Semistructured interviews addressed actions taken before seeking EC and delays once the need for EC was recognized. We purposefully interviewed a diverse sample of patient demographics and chief complaints. An interdisciplinary and multicultural research team conducted thematic analysis based on descriptive phenomenology, using the Framework method.” 

- You state in the title that a grounded theory approach was used, however that is a very specific iterative approach, that requires more details on how the approach was followed, vs an analysis based in phenomenology for example. Make sure you truly used a complete grounded theory approach, rather than utilizing certain elements based in grounded theory. Specify how the Framework Method helps to follow a grounded theory approach.

This is a very valid insight. While a true grounded theory approach was the intention, we used more pragmatic approach which does not fully adhere to grounded theory. The mention of grounded theory has been removed throughout.

Line 111: you justify the choice for the 12 hour time mark in your response letter, I suggest adding that here. It provides additional rigor to the methods.

This was added in lines 125-127: “≥12 hours was set as a benchmark based on advice from the Ugandan clinicians working in the ED due to their observation that patients arriving outside of this timeframe are subjectively noted to have poor outcomes attributable, at least in part, to delayed presentation.”

Line 114: information rich cases. Briefly state how you determined who’d have significant experience with challenges in EC access prior to having interviewed them. What were the characteristics the ED staff suggested you to include to meet this criteria?

This was added on lines 139-140: “These conversations often focused on challenges with identifying the need for EC, obtaining EC from local facilities, finance, or transport.” 

Line 129-131: I am curious if severe illness of the patient was a common reason for caregivers to refuse the interviews. I imagine this could be a very stressful and potentially challenging interview for caregivers and patients to have. Where any resources put in place, in case they needed support?

For exactly these reasons we did not approach caregivers when patients were severely ill. A description was added on lines 145-146: “Recruitment and interviews did not interfere with patients’ clinical care and were not conducted when the patient was severely ill but rather once they were stabilized, if at all.” 

Line 132-145: good description of the interview guide development. Was piloting of the questions performed? If not, what was the rationale?

Added on lines 173 -175: “The guide was initially piloted with community members not seeking EC followed by approximately ten ED patients or their caregivers.”

Line 146: briefly describe how the training in qualitative interview-based methods looked like. Didactics only, practice sessions, etc.

Added on lines 177-179: “After didactic training in qualitative interview-based research methods by AP, the head research assistant (NC), a resident of Rukungiri District and native Runyankore/Rukiga and fluent English speaker, obtained practical training, experience and feedback while piloting the interview guide.” 

Lines 172- 183: the data saturation statement is still not clear. I understand the rationale about adding more interviews to have a broader demographic profile, but it still sounds like saturation was reached after 25 interviews and not after 50. How exactly was saturation determined? Provide more details on the type of saturation (thematic vs data saturation)

Revised lines 204 -211 for clarity: After the first 25 interviews were conducted and analyzed, the data analysis team identified that no additional large-scale concepts of delay were emerging; thematic saturation was reached. The team convened to review subthemes across these first 25 interviews simultaneously and organize them, yielding the 4 themes described in Results. These themes and subthemes constituted a codebook, included as Appendix S2, facilitating coding of an additional 25 interviews completed to capture a broader sample of demographic groups and chief complaints across both dry and rainy seasons, given differences in EC accessibility and subsistence agriculture activities.

Line 194: I appreciate switching the Tables 1 and 2. Much clearer

Thank you for the useful suggestion. 

198: the distinction of “delay in ED presentation” and “Themes in Delay” is not very clear. Delay in ED presentation provides the quantitative explanation of delays, whereas themes in delay further explains and seeks to understands the delays.

Although you understood and stated the difference perfectly this is a very helpful clarification to make. This was added on lines 237-240: “Delay in ED presentation provides the quantitative explanation of delays, whereas themes in delay — presented below — seek to further understand and explain these delays in ED presentation.

220: A brief overview of themes and subthemes in table form or some visual could be helpful

A very helpful idea. Table 4. Themes and subthemes in delay identified was added on lines 244 – 245:

Table 4. Themes and subthemes in delay identified 

Theme Subthemes

Cultural Factors and Limited Knowledge of Emergency Signs and Initial Actions to Take • Difficultly recognizing “danger signs”

• Patient’s condition becoming severe before identifying the need for and/or seeking EC

• Perception of “cultural illness”

Use of Local Facilities for Stabilization and Advice Despite Perception of Inadequate Services • Limited first aid or home care

• Use of local facilities:

1. Assessable on foot or bicycle

2. For advice and symptom management

3. While gathering funds needed for EC

4. Despite perception of poor care at high cost

• Use of multiple facilities before seeking EC

• Perception of delays in referral to EC

Lack of Resources to Cover the Direct, Indirect, and Opportunity Costs of EC • Perception of EC as unaffordable

• Need for funds for transport prior to seeking EC

• Opportunity costs of EC

Inadequate Transportation Options, Especially at Night and in Inclement Weather • Inadequate vehicles available for the patient’s condition

• Dirt roads virtually unusable in heavy rain

• Safety concerns when traveling at night

239: Could you include the specific word used for “cultural illness” in the local dialects?

I discussed this with Charles Ndyamwijuka and the Ugandan members of our research team all of whom are native speakers. There is no one specific word for “cultural illness” these were identified through patient or caregivers’ descriptions of the presumed etiologies and treatment strategies.

259: Do you have information on what the maximum number of facilities visited before the ED presentation was?

Unfortunately, I do not. We queried types of facility and estimated costs but did not ask for a specific number of facilities. The use of multiple facilities was a somewhat unexpected result and was identified as multiple interviews were compared. 

288: I understand you mentioned in your response letter that it is difficult to provide context on how much a dollar sum is in the Ugandan context, specifically in this region, where a national net income would likely be an overestimate. But I think it is important to emphasize somehow, that 150$ is a significant sum and potentially catastrophic healthcare expenditure. Maybe you could add some small section in the description of the setting (lines 68-69)

This is included on lines 80-81: “The vast majority of inhabitants are subsistence farmers (78%), and most lack regular access to money, especially between harvest seasons [6].” 

324: change to past tense, check for consistency throughout

This was changed and the rest of the manuscript reviewed.

343: Discussion makes the objectives and planned next steps much clearer, positioning this paper in the ongoing research efforts. Good integration of the available literature.

Thanks!

Reviewer #2: (No Response)

Reviewer #3: I think you have addressed and covered the reasons for delay in careseeking of this area in Uganda well and got results that are in line with the Three Delays Model. These results are discussed thoroughly in the discussion as well as suggestions and thoughts on how to move forward are pointed out. With all studies there are limitations and I think you are aware of them and address them wisely.

---

## [Decision Letter · Decision Letter 2]

19 Jul 2022

Getting to the Emergency Department in Time: Interviews With Patients and Their Caregivers on the Challenges to Emergency Care Utilization in Rural Uganda

PONE-D-21-13731R2

Dear Dr. Pickering,

We’re pleased to inform you that your manuscript has been judged scientifically suitable for publication and will be formally accepted for publication once it meets all outstanding technical requirements.

Kind regards,

George Vousden

Staff Editor

PLOS ONE

Additional Editor Comments (optional):

Reviewers' comments:

Reviewer's Responses to Questions

**Comments to the Author**

1. If the authors have adequately addressed your comments raised in a previous round of review and you feel that this manuscript is now acceptable for publication, you may indicate that here to bypass the “Comments to the Author” section, enter your conflict of interest statement in the “Confidential to Editor” section, and submit your "Accept" recommendation.

Reviewer #2: All comments have been addressed

2. Is the manuscript technically sound, and do the data support the conclusions?

Reviewer #2: Yes

3. Has the statistical analysis been performed appropriately and rigorously? 

Reviewer #2: Yes

4. Have the authors made all data underlying the findings in their manuscript fully available?

Reviewer #2: Yes

5. Is the manuscript presented in an intelligible fashion and written in standard English?

Reviewer #2: Yes

6. Review Comments to the Author

Reviewer #2: The manuscript has been appropriately modified. authors have adequately addressed the suggestions from the reviewers.

7. PLOS authors have the option to publish the peer review history of their article (what does this mean?). If published, this will include your full peer review and any attached files.

Reviewer #2: No

---

## [Editor Report · Acceptance letter]

27 Jul 2022

PONE-D-21-13731R2 

Getting to the Emergency Department in Time: Interviews With Patients and Their Caregivers on the Challenges to Emergency Care Utilization in Rural Uganda 

Dear Dr. Pickering:

I'm pleased to inform you that your manuscript has been deemed suitable for publication in PLOS ONE. Congratulations! Your manuscript is now with our production department. 

Kind regards, 

on behalf of

Dr. George Vousden 

Staff Editor

PLOS ONE